# Norcantharidin Suppresses YD-15 Cell Invasion Through Inhibition of FAK/Paxillin and F-Actin Reorganization

**DOI:** 10.3390/molecules24101928

**Published:** 2019-05-19

**Authors:** Kyoung-Ok Hong, Chi-Hyun Ahn, In-Hyoung Yang, Jung-Min Han, Ji-Ae Shin, Sung-Dae Cho, Seong Doo Hong

**Affiliations:** Department of Oral Pathology, School of Dentistry and Dental Research Institute, Seoul National University, Seoul 03080, Korea; hongko95@snu.sc.kr (K.-O.H.); chihyun610@snu.ac.kr (C.-H.A.); inhyoung3@naver.com (I.-H.Y.); 2017-20655@snu.sc.kr (J.-M.H.); sky21sm@snu.ac.kr (J.-A.S.)

**Keywords:** mucoepidermoid carcinoma, norcantharidin, invasion, F-actin reorganization, focal adhesion kinase, paxillin

## Abstract

Norcantharidin (NCTD), a demethylated derivative of cantharidin, has been reported to exhibit activity against various types of cancers. However, the anti-invasive effects of NCTD and its molecular mechanism in human mucoepidermoid carcinoma (MEC) remain incompletely elucidated. Clonogenic, wound healing, invasion, zymography, western blotting and immunocytochemistry assays were performed in YD-15 cells to investigate the anti-invasive effect of NCTD and its molecular mechanism of action. The inhibitory effects of NCTD on invasiveness were compared with those of a novel focal adhesion kinase (FAK) kinase inhibitor, PF-562271. NCTD markedly suppressed the colony formation, migration, and invasion of YD-15 cells as well as the activities of MMP-2 and MMP-9. It disrupted F-actin reorganization through suppressing the FAK/Paxillin axis. Moreover, NCTD exhibited a powerful anti-invasive effect compared with that of PF-562271 in YD-15 cells. Collectively, these results suggest that NCTD has a potential anti-invasive activity against YD-15 cells. This study may clarify the impact of NCTD on migration and invasion of human MEC cells.

## 1. Introduction

Mucoepidermoid carcinoma (MEC) is the most common malignant tumor of the major salivary glands and histologically comprises variable mixtures of mucin-secreting, intermediated, and epidermoid cells [1,2]. MEC is characterized by ready penetration of surrounding tissues and is likely to recur easily [3]. As with other solid tumors, traditional methods for treating MEC, such as chemotherapy, can cause strong side effects and drug resistance [4]. Therefore, it is necessary to continue searching for novel, efficient, and less toxic cancer therapeutic drug candidates.

Invasion and metastasis belong to the six cancer biological capability hallmarks which were described by Hanahan and Weinberg, and represent distinct traits between benign and malignant tumors [5,6]. To metastasize, cancer cells require a mechanical event that involves adhesion, shape change, movement, and force generation; only then can they finally undergo infiltration through the extracellular matrix (ECM) in order to form new tumors at another site in the human body [7]. With remodeling of the ECM, reorganization of the actin cytoskeleton is required for invasion and metastasis of cancer and is regulated by a variety of proteins, including actin regulatory proteins, adhesion molecules, signaling proteins, and matrix degradation enzymes [8,9,10]. Focal adhesion kinase (FAK) is a non-receptor tyrosine kinase that acts as an important mediator of integrin-mediated signaling between cells and the ECM. It serves as a scaffolding protein for the binding sites of multiple oncogenic tyrosine kinases and regulates paxillin function through phosphorylation to induce diverse cellular processes, including adhesion, migration, invasion, and metastasis [11,12]. Indeed, FAK is found at an elevated level in most human cancers, particularly when the cancer is transformed into highly invasive metastases [13,14], suggesting that FAK is a key signaling molecule for an effective cancer therapeutic strategy. However, the functional significance of FAK activity in MEC has not yet been fully understood.

Natural products such as plant and animal-derived drugs play an increasingly important role in cancer treatment due to their low side-effects and high efficacy [15,16]. Recently, NCTD, a synthetic demethylated analog of cantharidin isolated from *Mylabris phalerata* Pallas (blister beetle), has been reported to have potent anticancer activity, including the to inhibit capacity tumor invasiveness and metastasis in colorectal and breast cancer cells [17,18]. NCTD is known as an inhibitor of protein Ser/Thr phosphatases (PPP), which regulate various important physiological and pathological processes, such as cell cycle and proliferation [19]. Despite several studies on the biological activities of NCTD, its anti-proliferative and anti-invasive potential are not yet defined. In the present study, we investigated the anti-invasive potential of NCTD in the YD-15 human MEC cell line and compared its anti-cancer function with that of PF562271, a potent FAK inhibitor.

## 2. Results

### 2.1. Low Concentrations of NCTD Affect Survival without Apoptosis in the YD-15 Cell Line

To examine the inhibitory effect of NCTD on cell growth, a Trypan Blue exclusion assay was performed on YD-15 cells after exposure to various concentrations of NCTD or 0.1% DMSO (vehicle control) for 24 h. Low concentrations (2.5–10 μM) of NCTD did not inhibit cell viability (Figure 1B), but high concentrations (20–40 μM) of NCTD significantly reduced cell viability (Appendix A). 

To further investigate the effects of NCTD on the survival and proliferation of YD-15 cells, we performed a 10-day clonogenic assay. As shown in Figure 1C,D, 10 μM NCTD exhibited significant inhibition of colony formation. Western blotting and flow cytometric analyses with Annexin V/PI double staining were performed to confirm whether the inhibition of colony formation by NCTD induces apoptosis. There was no change in YD-15 cells treated with low concentrations of NCTD (data not shown), but high concentrations of NCTD significantly increased cleavages of caspase 3 and PARP (Appendix A). In addition, the number of annexin V-positive cells in 40 μM NCTD-treated YD-15 cells increased (Appendix A). These results suggest that low concentrations of NCTD affect survival without causing apoptotic cell death in YD-15 cells. Thus, we used NCTD of 10 μM or less for all subsequent experiments.

### 2.2. NCTD Represses YD-15 Cell Migration and Invasion through Down-Regulation of MMP-2 and MMP-9 Activity

A wound healing assay was performed to measure the effects of NCTD on cell migration. YD-15 cells showed significantly reduced wound closure levels after NCTD treatment in a concentration-dependent manner compared with vehicle control (Figure 2A,B). 

The effect of NCTD on cell invasion was also investigated using a transwell chamber pre-coated with matrigel. Cells treated with NCTD showed a significant decrease in cell invasion similar to those obtained from the wound healing assay (Figure 2C,D). In order to evaluate the mechanism underlying their inhibitory action on migration and invasion, we measured the activities of MMP-2 and MMP-9 involved in tumor cell migration and invasion by gelatin zymography [7]. The results showed that the activities of both MMP-2 and MMP-9 were significantly decreased by NCTD treatment (Figure 2E,F). These results suggest that NCTD regulates MMP-2 and MMP-9 to inhibit migration and invasion of the human YD-15 cell line.

### 2.3. NCTD Inhibits F-Actin Reorganization of YD-15 Cells through Inactivation of FAK and Paxillin

In order to determine whether cytoskeletal reorganization signaling pathways are implicated in the anti-invasive effects of NCTD, immunofluorescence analysis was performed on fixed cells. Actin stress fiber and filopodia were largely abolished and disorganized by NCTD treatment compared with the vehicle control (Figure 3A). Next, FAK, Src, and paxillin, the most common signaling components of F-actin reorganization, were evaluated using western blotting. Activation of FAK (Tyr 397) and paxillin (Tyr 118) was strongly attenuated by NCTD treatment, but NCTD exhibited no apparent effect on Src activation (Try 416) (Figure 3B,C). These data suggested that NCTD may act as an inhibitor of the FAK/paxillin signaling axis to promote F-actin reorganization in YD-15 cells.

### 2.4. NCTD Inhibits the Invasiveness of YD-15 Cells More Potently than PF562271

PF562271 is a potent small molecule inhibitor of FAK and has been reported to inhibit the proliferation of cancer cells [20,21]. Thus, we first evaluated the effects of PF562271 on the expression levels of p-FAK, Src, and paxillin in YD-15 cells before evaluating the anti-invasive effect of NCTD. The results showed that PF562271 significantly inactivated FAK (Tyr 397) and paxillin (Tyr 118) similar to NCTD (Figure 4). Next, the anti-invasive effect of NCTD was compared with PF562271 in YD-15 cells. NCTD dramatically decreased wound closure speed (Figure 5A,B) and led to inhibition of invasion (Figure 5C). 

NCTD also induced F-actin disorganization and decreased MMP-2 and MMP-9 activity according to immunofluorescence analysis and gelatin zymography (Figure 5D–F). However, treatment with PF562271 exhibited no apparent effects on cell migration and invasion, inhibition of MMP-2 and MMP-9 activity, or disorganized stress fibers. Collectively, these data suggested that NCTD could inhibit the migration and invasion in YD-15 cells more potently than PF562271.

## 3. Discussion

The invasion and metastasis of cancer cells are closely related to cancer recurrence and death in patients with oral cancer [22]. Although the introduction of synthesized chemotherapeutic agents for the treatment of human cancers with invasion and metastasis has been implicated as one of the notable advances in cancer research over the last three decades, some of them were disappointing because of unwanted and harmful side effects [5,23,24]. Thus, the search for therapeutic natural products with fewer side effects, such as NCTD, is an increasingly important area for chemotherapeutic drug discovery [15,25]. Although there has been much research on the efficacy of NCTD-induced apoptosis in cancer cells, little is known about the potential activity of NCTD against migration and invasion [19,26]. Herein, the effect of NCTD on invasiveness of YD-15 cells was investigated. We found that NCTD dramatically inhibited YD-15 cell migration and invasion in a concentration-dependent manner via down-regulation of MMP-2 and MMP-9 activity. These results are consistent with other previous studies using different types of cancer, including gallbladder and liver cancers [27,28] and suggest that NCTD may have anti-invasive activity by inhibiting the activity of MMPs in human MEC cells.

Peng et al. [29] reported that very high concentration of NCTD (60 μM) could suppress MMP-9 expression levels as well as the reorganization of F-actin in colon cancer. In our current study, the use of only 5 or 10 μM NCTD significantly abolished F-actin reorganization and filopodia-like protrusions in YD-15 cells. These findings indicate that NCTD has much greater potential for MEC than colon cancer. FAK and Src has been shown to play a crucial role in the production of MMPs and F-actin reorganization, leading to tumor invasion [12,30]. Especially, Src is a signaling protein that regulates cytoskeletal dynamics and cell motility [31]. Phosphorylated Tyr527 residue of Src interacts and binds with the SH2 domain, keeping in the inactive conformation, but the Tyr416 residue is essential for Src activation [32]. However, no modification of Src was seen in response to NCTD stimulation, while NCTD dephosphorylated FAK (Y397). These results are inconsistent with the findings of Hsia et al. who showed that the treatment of platelets with NCTD inhibited phosphorylation of both Src and FAK [33]. Collectively, NCTD has the potential to inhibit F-actin reorganization in human MEC cells, and the underlying mechanism may be related to the FAK/Paxillin signaling axis.

To further clarify whether the FAK/paxillin signaling axis may be involved in the anti-invasive activity of NCTD, we compared its activity with a potent, selective FAK inhibitor, PF562271 [34]. PF562271 was previously shown to display inhibitory effects on FAK phosphorylation and migration in human umbilical vascular endothelial cells, Ewing sarcoma, and human keratinocytes [35,36,37]. PF562271 attenuated the phosphorylation expression levels of FAK and paxillin. However, unexpectedly, PF562271 weakly suppressed only the invasiveness of YD-15 cells in the transwell invasion assay and had no effect on migration, F-action reorganization, or MMPS activity of YD-15 cells. Contrary to PF562271, NCTD completely inhibited the invasion of YD-15 cells. Vinculin is localized in focal adhesion as well as cell-adherence junctions (AJ) and is phosphorylated on residues Y100 and Y1065 by members of the Src family before binding to F-actin [38]. Recently, Bays et al., reported that phosphorylation of vinculin at Y822 can increase when forces are applied to cell–cell junctions (E-cadherin), which indicates the regulatory function of Y822 occurs by distribution of the phosphorylation of some focal adhesion proteins including paxillin and FAK [39]. One possible explanation for the different responses of NCTD and PF562271 is that p-vinculin can be affected differently by the two chemicals. Although future studies on the current issue are recommended, these finding clearly demonstrate more potent efficacy of NCTD on F-actin disorganization compared with PF562271.

In conclusion, the present in vitro study indicates that NCTD may have an anti-invasive potential for human MEC through a novel mechanism involving inhibition of F-actin binding-related proteins. This provides tremendous support for the role of NCTD as an attractive anticancer drug candidate against human MEC.

## 4. Materials and Methods

### 4.1. Cell Culture and Chemical Treatment

The YD-15 cell line was obtained from Yonsei University (Seoul, Korea) and was cultured in RPMI-1640 medium supplemented with 10% fetal bovine serum (FBS) and antibiotics at 37 °C in a 5% CO_2_ incubator. NCTD (Figure 1A) was purchased from Sigma-Aldrich Chemical Co. (St. Louis, MO, USA) and used for treatment at various concentrations ranging from 0 to 40 μM for 24 h. PF562271 (a focal adhesion protein-tyrosine kinase inhibitor) was purchased from Selleckchem (Houston, TX, USA). Each chemical was dissolved in dimethyl sulfoxide (DMSO), aliquoted, and stored at −20 °C. All treatments were performed after cells reached 50–60% confluence.

### 4.2. Trypan Blue Exclusion Assay

YD-15 cells were treated with different concentrations of NCTD for 24 h, and cell viability was measured using trypan blue staining (Gibco, Paisley, UK). Cells were stained with 0.4% trypan blue solution, and viable cells were counted using a hemocytometer.

### 4.3. Clonogenic Formation Assay

Assays were performed as previously described [40]. Briefly, YD-15 cells were seeded into a 12-well culture plate with 1000 cells per well. After 18 h, NCTD was added to the wells, and the plate was re-incubated. The medium was replaced with fresh medium containing NCTD every three days for 1 week. The colonies were then stained with 1% crystal violet and counted.

### 4.4. Scratch Wound Healing Assay

When cells reached 90–100% confluence in 6-well culture plates, YD-15 cells were slowly scratched using a sterile plastic pipette tips across the center of the well. The cells were gently washed twice in warm medium to remove the detached cells, immediately imaged (0 h), and treated with DMSO, NCTD (5 and 10 μM), or PF562271 (0.5 μM) for 24 h. Images of wound gaps were then time-dependently acquired using a CKX53 microscope (Olympus, Tokyo, Japan). The wound closure areas were quantified as a percentage of cell migration into the wound with respect to the clear area at 24 h. The migration area was measured using the ImageJ software (Version 1.51k; NIH, Bethesda, MD, USA).

### 4.5. Matrigel Invasion Assay

YD-15 cells (5 × 10^4^ cells/well) were treated with NCTD or PF562271 for 24 h on pre-coated 24-well inserts (8 μm pore size) with Matrigel (BD Bioscience, Bedford, MA, USA). The lower chamber was filled with medium containing 10% FBS as a chemoattractant. The transwell chamber was incubated at 37 °C with 5% CO_2_ for 24 h. The cells on the upper surface of the filter were then wiped off with a cotton swab, and the filter was removed from the chamber and stained with hematoxylin and eosin. The number of cells on lower chamber were counted under a light microscope (Leica DM5000B; Leica Microsystems, Wetzlar, Germany). For each assay, 10 different microscopic fields (×100 magnification) were randomly chosen.

### 4.6. Gelatin Zymography

YD-15 cells were treated in serum-free medium with NCTD or PF562271 for 72 h. The conditioned medium was collected and concentrated using Amicon Ultra Centrifugal Filter Units (Millipore, Billerica, MA, USA). To analyze MMP-2 and MMP-9 activities, equal amounts of protein (30 μg) were loaded onto a gelatin-containing gel (8% acrylamide gel containing 1.5 mg/mL gelatin) and separated by electrophoresis. The gel was renatured in 2.5% Tween-20 solution with gentle agitation for 30 min at room temperature (RT), developed overnight at 37 °C in zymogram incubation buffer (50 mM Tris–HCl (pH 7.6) and 5 mM CaCl_2_), and stained with Coomassie Brilliant Blue R250 (Bio-Rad Laboratories, Hercules, CA, USA). The gel was then de-stained with a solution of 50% methanol and 10% acetic acid until the part of the membrane degraded by MMP-2 and MMP-9 became clear. The density of the clear bands was determined using ImageJ software.

### 4.7. F-Actin Staining Using TRITC-Conjugated Phalloidin

YD-15 cells were seeded on coverslips and treated with DMSO, NCTD, or PF562271 for 24 h. The cells were fixed and permeabilized with Cytofix/Cytoperm (BD Bioscience) for 40 min at 4 °C and incubated with TRITC-labelled phalloidin for 40 min, washed with PBS containing 0.1% BSA, and then reacted with 5 mg/mL 4,6-diamidino-2-phenylindole (DAPI; Sigma) for nuclei staining. After washing, the samples were rinsed with PBS and mounted with a mounting media. Immunofluorescence images were obtained using LSM700 confocal laser scanning microscope (Carl Zeiss, Oberkochen, Germany).

### 4.8. Western Blot Analysis

Proteins were extracted from cell pellets by homogenization with RIPA buffer (EMD Millipore, Billerica, CA, USA), and the protein concentration of each sample was measured using a DC Protein Assay Kit (Bio-Rad Laboratories). After normalization, equal amounts of protein were separated by sodium dodecyl sulfate-polyacrylamide gel electrophoresis and transferred to immunoblot polyvinylidene difluoride membranes (Pall Corporation, Port Washington, NY, USA). The membranes were blocked with 5% skim milk at RT for 2 h, incubated with the specific primary antibodies, and probed with corresponding horseradish peroxidase-conjugated secondary antibodies (GTX213110 for anti-Rabbit and GTX213111 for anti-mouse). Rabbit anti-human polyclonal antibodies against cleaved poly (ADP-ribose) polymerase (PARP; 1:3000; #9541), cleaved caspase-3 (1:1000; #9664), p-FAK(Y397; 1:1000; #3283), FAK (1:1000; #32853), p-Paxillin (Y118; 1:1000; #2541), Paxillin (1:1000; #2542), p-Src (Y416; 1:1000; #6943), and Src (1:1000; #2109) were purchased from Cell Signaling Technology (Charlottesville, VA, USA). Mouse anti-human monoclonal antibodies against β-actin (1:3000; SC-47778) and GAPDH (1:3000; ab9484) were obtained from Santa Cruz Biotechnology (Santa Cruz, CA, USA) and Thermo Fisher Scientific (Rockford, IL, USA), respectively. Immunoreactive proteins were detected by SuperSignal West Pico Chemiluminescent Substrate (sc-2048; Santa Cruz Biotechnology). The immunoreactive bands were visualized using ImageQuant LAS 500 (GE Healthcare Life Sciences, Pittsburgh, PA, USA). The densitometric analysis of western blotting was quantified using ImageJ software (NIH).

### 4.9. Statistical Analysis

Data are expressed as mean ± S.D. One-way ANOVAs analysis was applied to determine the significance of differences between the control and treatment groups using SPSS v22 (SPSS, Chicago, IL, USA); values of *p* < 0.05 were considered statistically significant (*).

## Figures and Tables

**Figure 1 molecules-24-01928-f001:**
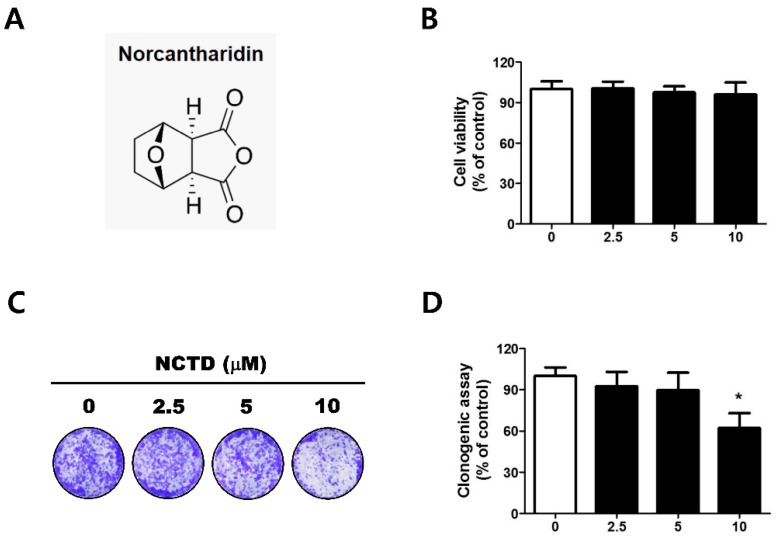
NCTD inhibits colony formation of YD-15 cells. (**A**) Chemical structure of NCTD. (**B**) Trypan blue exclusion assay for cell viability after the treatment of NCTD (2.5, 5, and 10 μM) for 24 h. (**C**,**D**) Clonogenic assay for colony formation of YD-15 cells. Graphs show the mean ± SD of triplicate experiments and significance compared with the vehicle control (* *p* < 0.05).

**Figure 2 molecules-24-01928-f002:**
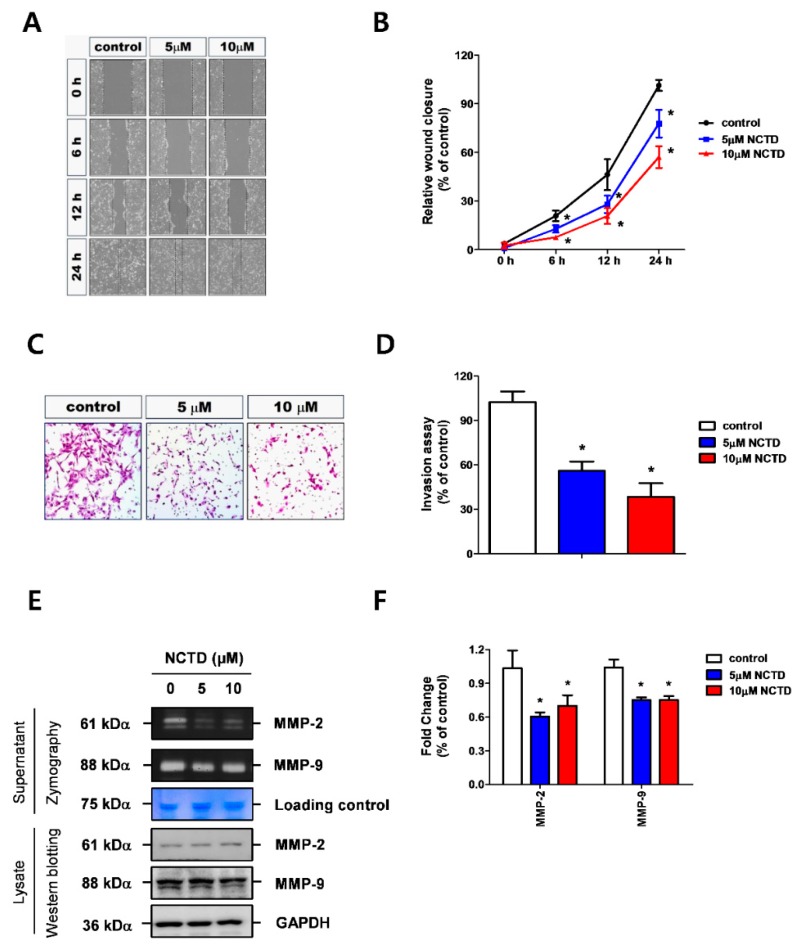
NCTD inhibits the migration and invasion of YD-15 cells. (**A**,**B**) Wound-healing assay of cell migration after treatment of NCTD (5 and 10 μM) in YD-15 cells for 24 h. The dotted lines delineate clear zones at various time-points. (**C**,**D**) Transwell assay for cell invasive potential after treatment with NCTD in YD-15 cells. (**E**,**F**) Effects of NCTD on MMP activity using gelatin zymography and western blotting. Graphs show the mean ± SD of triplicate experiments and significance compared with the vehicle control (* *p* < 0.05).

**Figure 3 molecules-24-01928-f003:**
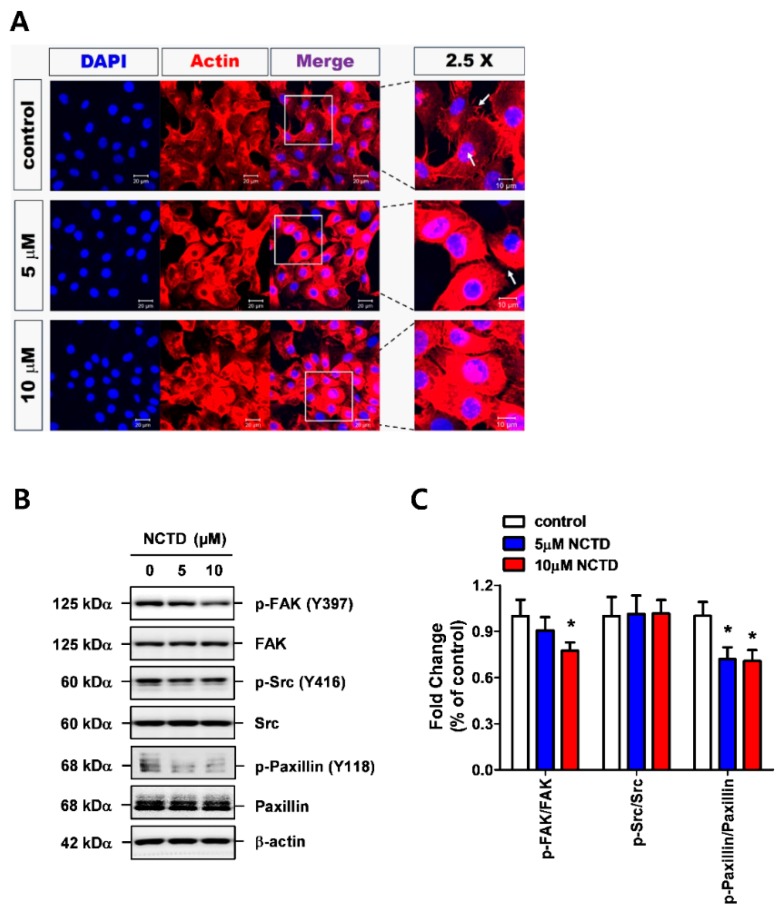
NCTD induces F-actin reorganization and activates FAK/Src/paxillin signaling. (**A**) Confocal images of YD-15 cells in the absence and presence of NCTD (5 and 10 μM). Actin staining by fluorescent phalloidin (red) and nuclear staining by DAPI (blue). The white arrow indicates actin with elongated filopodia and stress fibers. The images are representative of three independent experiments. (**B**,**C**) Western blot analysis of the phosphorylation states of FAK/Src/paxillin signaling. Total protein expressions of FAK, Src, and paxillin were used for normalization. Graphs show the mean ± SD of triplicate experiments and significance compared with the vehicle control (* *p* < 0.05).

**Figure 4 molecules-24-01928-f004:**
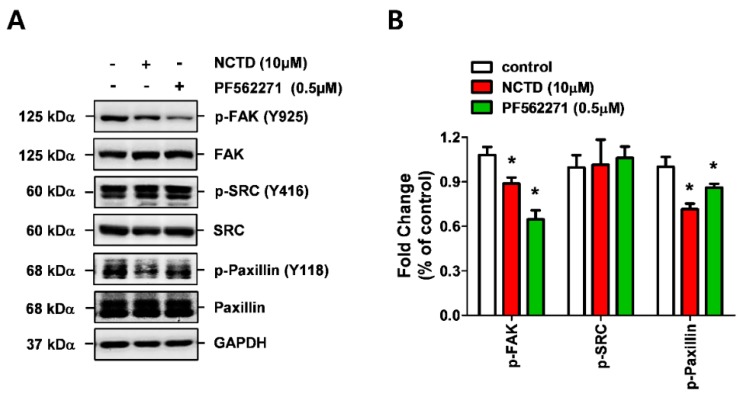
Comparison of the activation of FAK/Src/paxillin signaling between NCTD and PF562271 in YD-15 cells. (**A**,**B**) Western blot analysis of the phosphorylation of FAK/Src/paxillin signaling after the treatment with NCTD (10 μM) or PF562271 (0.5 μM) for 24 h. Graphs show the mean ± SD of triplicate experiments and significance compared with the vehicle control (* *p* < 0.05).

**Figure 5 molecules-24-01928-f005:**
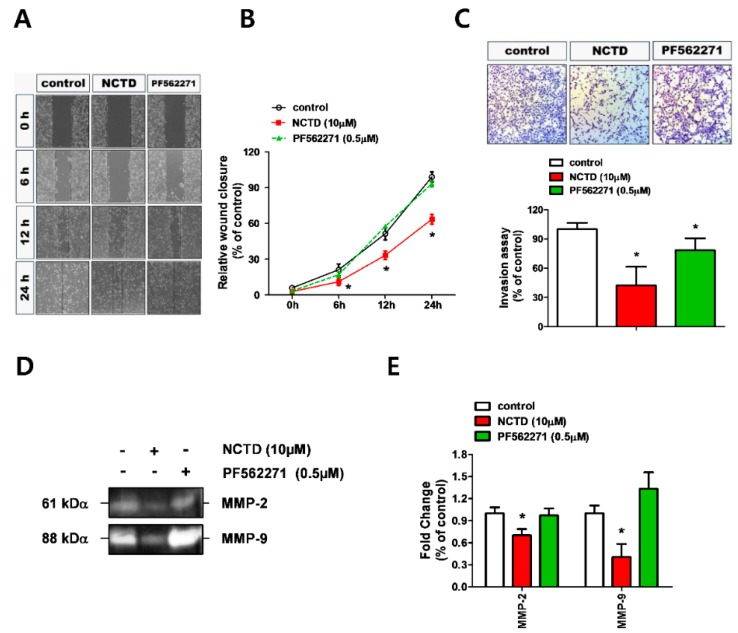
Comparison of the anti-invasive efficacy between NCTD and PF562271 in YD-15 cells. (**A**,**B**) Wound-healing assay of cell migration after treatment with NCTD or PF562271. (**C**) Transwell assay for the cell invasive potential after treatment with NCTD or PF562271 in YD-15 cells. (**D**,**E**) Effects of NCTD or PF562271 on MMP activity using gelatin zymography. (**F**) Confocal images of YD-15 cells after treatment with NCTD or PF562271. Graphs show the mean ± SD of triplicate experiments and significance compared with the vehicle control (* *p* < 0.05).

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
