# Peer review of "Norcantharidin Suppresses YD-15 Cell Invasion Through Inhibition of FAK/Paxillin and F-Actin Reorganization"

_molecules, 2019, doi:10.3390/molecules24101928_

Round 1

Reviewer 1 Report

The manuscript is interesting and worthy of publication in the Molecules. The authors showed that NCTD suppresses the invasiveness of YD-15 cells through F-actin changes and depending on the FAK/Paxillin axis. 

I suggest only to improve and enlarge fluorescent micrographs.

Author Response

Reviewer 1
The manuscript is interesting and worthy of publication in the Molecules. The authors showed that NCTD suppresses the invasiveness of YD-15 cells through F-actin changes and depending on the FAK/Paxillin axis. I suggest only to improve and enlarge fluorescent micrographs.

è We appreciate your interest on our manuscript. As you suggested, we improved fluorescent micrographs for Figure 3A and 5F. [Lines 143, 178]

Reviewer 2 Report

Hong et al. report on the effects of an anticancer small molecule, norcantharidin (NCTD), on cell culture assays related to cancer cell (YD-15 cell line) invasiveness and metastasis.  NCTD inhibits cell motility, with evidence pointing to effects on the FAK signaling pathway and regulation of the actin cytoskeleton.  The results suggest that NCTD (or related compounds) might be useful for treating metastatic epidermal (and perhaps other) cancers.

Major

With regard to the antibodies against phosphorylated proteins, the authors should include any information about which residues are reported on by the antibodies, and any information about the robustness of the assay when multiple residues can be phosphorylated in a protein.  Furthermore, the authors should discuss the implications of results reporting phosphorylation status when phosphorylation of various residues in one protein (e.g., SRC) can increase, decrease or have no effect on activity (Roskoski 2005 BBRC https://doi.org/10.1016/j.bbrc.2005.03.012).

The authors should incorporate into their discussion a comparison between their focus on phosphorylation of FAK in contrast to (or perhaps in combination with) Shen & Schaller’s suggestion that “subcellular localization is the major determinant of FAK function.”  (2017 MBoC https://doi.org/10.1091/mbc.10.8.2507)

Minor

Overall, the writing is quite good.  That said, there are some portions of the text that could use editing.  Below is a sample; it is not intended to comprehensively list every instance.  The authors should thoroughly edit the entire manuscript for language.

Abstract lines 21-22: This sentence in the Abstract says that “NCTD markedly suppressed…the decreases in MMP-2 and 9 activity.”  However Figure 5D,E shows that NCTD reduced the activities of MMP-2 and 9. It did not suppress a decrease in enzyme activity.

Intro line 41: “…in other site of…” should probably be something along the lines of either “…in another site in…” or “…in other sites of…”

Intro line 41: could consider replacing “…is highly associated with…” with “…is required for…”

Intro line 55: shouldn’t the genus name “mylabris” be capitalized and italicized?

Results line 84: “…repress…” should be “…represses…” and “…cells…” should be “…cell…”

Results lines 92-95 and Figure 2 legend line 102: Results “…we measured the activities of MMP-2 and MMP-9 involved in tumor cell migration and invasion by degrading extracellular matricies [7]. The results showed that the activities of both MMP-2 and MMP-9…” and the legend for Figure 2E,F “[e]ffects of NCTD on MMP activity” should clarify that activity was assessed by gelatin zymography.  The reader shouldn’t have to go to the methods section to find this out.  Also note that spelling of “matricies” needs to be corrected.

Figure 2 legend lines 100-101: “…dotted line represents a clear zone…” should probably be something along the lines of “…dotted lines delineate clear zones…”

Results lines 106-107 and legend for Figure 3A: clarify whether immunofluorescence analysis was performed on fixed cells.

Results lines 130-131 and Figure 5 legend line 146: Results “NCTD…decreased MMP-2 and MMP-9 activity…” and the legend for Figure 5D,E “[e]ffects of NCTD or PF562271 on MMP activity” should clarify that activity was assessed by gelatin zymography.  As noted above, the reader shouldn’t have to go to the methods section to find this out.

Discussion: the long paragraph starting on line 164 could probably be made more readable if broken up into two paragraphs.

Author Response

Reviewer 2

Major

With regard to the antibodies against phosphorylated proteins, the authors should include any information about which residues are reported on by the antibodies, and any information about the robustness of the assay when multiple residues can be phosphorylated in a protein. 

è We appreciate your valuable comments. As you suggested, we added the information about the phosphorylation residues for each kinase to the Materials and Methods and checked it again to make it sure. [Lines 323-324]

à “Rabbit anti-human polyclonal antibodies against cleaved poly (ADP-ribose) polymerase (PARP; 1:3000; #9541), cleaved caspase-3 (1:1000; #9664), p-FAK (Y397; 1:1000; #3283), FAK (1:1000; #3285), p-Paxillin (Y118; 1:1000; #2541), Paxillin (1:1000; #2542), p-SRC (Y416;1:1000; #6943), and SRC (1:1000; #2109) were purchased from Cell Signaling Technology (Charlottesville, VA, USA).”

Furthermore, the authors should discuss the implications of results reporting phosphorylation status when phosphorylation of various residues in one protein (e.g., SRC) can increase, decrease or have no effect on activity (Roskoski 2005 BBRC https://doi.org/10.1016/j.bbrc.2005.03.012).

è As you mentioned, we studied the implications of results reporting phosphorylation status. Src is a signaling protein that regulates cytoskeletal dynamics and cell motility (Ref. 1). Phosphorylated Tyr527 residue of Src interacts and binds with the SH2 domain, keeping in the inactive conformation, but phosphorylation of a key Tyr416 residue in the activation loop is essential for Src activation to perform cellular function (Ref. 2). Thus, we focused on the phosphorylation of Tyr416 residue of Src for the present study. 

è Based on the above information, we discussed this issue in the ‘Discussion’ section. [Lines 207-212]

FAK and Src has been shown to play a crucial role in the production of MMPs and F-actin reorganization, leading to tumor invasion [12, 29]. Especially, Src is a signaling protein that regulates cytoskeletal dynamics and cell motility (Ref). Phosphorylated Tyr527 residue of Src interacts and binds with the SH2 domain, keeping in the inactive conformation, but Tyr416 residue is essential for Src activation (Ref). However, no modification of Src was seen in response to NCTD stimulation, while NCTD dephosphorylated FAK (Y397).

Ref.1 Geiger, B.; Bershadsky, A.; Pankov, R.; Yamada, K. M., Transmembrane crosstalk between the extracellular matrix--cytoskeleton crosstalk. Nat Rev Mol Cell Biol 2001, 2, (11), 793-805.

Ref.2 Roskoski, R., Jr., Src kinase regulation by phosphorylation and dephosphorylation. Biochem Biophys Res Commun 2005, 331, (1), 1-14.

The authors should incorporate into their discussion a comparison between their focus on phosphorylation of FAK in contrast to (or perhaps in combination with) Shen & Schaller’s suggestion that “subcellular localization is the major determinant of FAK function.”  (2017 MBoC https://doi.org/10.1091/mbc.10.8.2507)

è Thank you for your kind comments and your comment is very important one. In Shen & Schaller’s a mutational strategy, FAK mutant is unable to induce phosphorylation of Paxillin to subcellular localization for focal adhesion (Ref.3). Thus Tyr 397 residue of FAK is necessary for both integrin-induced tyrosine phosphorylation of FAK and induction of tyrosine phosphorylation of substrates like Paxillin, focal adhesion-associated protein. Consequently, it can lead to cellular adhesion, migration and invasion via phosphorylation. Prior to your comments, we did focus on the phosphorylation levels of FAK and Paxillin. So, we didn’t check the subcellular localization of FAK and Paxillin using immunofluorescence staining. However, we determined that NCTD significantly abolished F-actin reorganization and filopodia-like protrusion using actin-phalloidin fluorescent staining in YD-15 cells (Figure 3A and 5F).

Ref.3 Shen, Y.; Schaller, M. D., Focal adhesion targeting: the critical determinant of FAK regulation and substrate phosphorylation. Mol Biol Cell 1999, 10, (8), 2507-18.

Minor

Overall, the writing is quite good. That said, there are some portions of the text that could use editing.  Below is a sample; it is not intended to comprehensively list every instance.  The authors should thoroughly edit the entire manuscript for language.

è Thank you for your kind and thoughtful comments. We have proofread the part and replaced it with correct information in point-by-point response to the reviewers’ comments.

Abstract lines 21-22: This sentence in the Abstract says that “NCTD markedly suppressed…the decreases in MMP-2 and 9 activity.”  However, Figure 5D, E shows that NCTD reduced the activities of MMP-2 and 9. It did not suppress a decrease in enzyme activity

àNCTD markedly suppressed the colony formation, migration, and invasion of YD-15 cells as well as the activities of MMP-2 and MMP-9.” [Abstract, line 22]

Intro line 41: “…in other site of…” should probably be something along the lines of either “…in another site in…” or “…in other sites of…”

àTo metastasize, cancer cells require a mechanical event that involves adhesion, shape change, movement, and force generation; only then can they finally undergo infiltration through the extracellular matrix (ECM) in order to form new tumors in another site in the human body” [Line 41]

Intro line 41: could consider replacing “…is highly associated with…” with “…is required for…”

à “With remodeling of the ECM, reorganization of the actin cytoskeleton is required for invasion and metastasis of cancer and is regulated by a variety of proteins, including actin regulatory proteins, adhesion molecules, signaling proteins, and matrix degradation enzymes [8-10].” [Line 42]

Intro line 55: shouldn’t the genus name “mylabris” be capitalized and italicized?

à “Recently, NCTD, a synthetic demethylated analog of cantharidin isolated from Mylabris phalerata Pallas (blister beetles), has been reported to have potent anticancer activity including inhibiting tumor invasive capacity and metastasis in colorectal and breast cancer cells.” [Line 60]

Results line 84: “…repress…” should be “…represses…” and “…cells…” should be “…cell…”

à “2.2. NCTD represses YD-15 cell migration and invasion through down-regulation of MMP-2 and MMP-9 activity” [Line 91]

Results lines 92-95 and Figure 2 legend line 102: Results “…we measured the activities of MMP-2 and MMP-9 involved in tumor cell migration and invasion by degrading extracellular matricies [7]. The results showed that the activities of both MMP-2 and MMP-9…” and the legend for Figure 2E,F “[e]ffects of NCTD on MMP activity” should clarify that activity was assessed by gelatin zymography.  The reader shouldn’t have to go to the methods section to find this out.  Also note that spelling of “matricies” needs to be corrected.

à “In order to evaluate the mechanism underlying their inhibitory action on migration and invasion, we measured the activities of MMP-2 and MMP-9 involved in tumor cell migration and invasion by gelatin zymography [7].” [Line 100]

à “(E-F) Effects of NCTD on MMP activity using gelatin zymography and western blotting.” [Lines 110-111]

Figure 2 legend lines 100-101: “…dotted line represents a clear zone…” should probably be something along the lines of “…dotted lines delineate clear zones…”

àThe dotted lines delineate clear zones at various time-points.” [Lines 108-109]

Results lines 106-107 and legend for Figure 3A: clarify whether immunofluorescence analysis was performed on fixed cells.

à “In order to determine whether cytoskeletal reorganization signaling pathways are implicated in the anti-invasive effects of NCTD, immunofluorescence analysis was performed on fixed cells.” [Line 115]

Results lines 130-131 and Figure 5 legend line 146: Results “NCTD…decreased MMP-2 and MMP-9 activity…” and the legend for Figure 5D,E “[e]ffects of NCTD or PF562271 on MMP activity” should clarify that activity was assessed by gelatin zymography.  As noted above, the reader shouldn’t have to go to the methods section to find this out.

àNCTD also visualized F-actin disorganization and decreased MMP-2 and MMP-9 activity using immunofluorescence analysis and gelatin zymography (Figures 5D-5F).” [Line 160]

à “(D-E) Effects of NCTD or PF562271 on MMP activity using gelatin zymography.” [Line 184]

Discussion: the long paragraph starting on line 164 could probably be made more readable if broken up into two paragraphs

è As you suggested, we divided into two paragraphs at line 216 for better readability.

Reviewer 3 Report

Cantharidin (and norcantharidin) are reported anti-cancer agents that potentially can escape multidrug resistance and therefore might be of interest as to establish therapeutic schemes alone or in combination with other better defined chemotherapeutic agents.

The authors conducted a research plan in line to similar studies and I only wish to note some seemingly conflicting results that the authors may wish to address in their response and the manuscript:

The dominant mechanism of CTD/NCTD is inhibitory activity against PP1 and PP2A protein phosphatases and this is largely ignored in the manuscript

Given the above I find it quite strange that phosphorylation of FAK (and even more of paxillin) is downregulated and that in this case is less dose sensitive than wound closure

ERK activation is a predominant feature of MEK aggresion and the authors should address the effects of NCTD on phospho-ERK levels

Activities of matrix metalloproteases 2 and 9 are differentially regulated but it will be important to show what is the trend of their overall amounts via westerns blotting. Additionally Fig. 2E lacks loading controls

The very big difference in the inhibitory concentrations between NCTD and PF562271 further signifies that NCTD effects on FAK and paxillin phosphorylation maybe very indirect and thus the evaluation of NCTD without concrete knowledge of its target(s) is somewhat inconclusive

Phalloidin staining of actin is not immunofluorescence so this has to be changed in title 4.7 and any where else it is relevant

densitometric analysis of western blots should consider the ratios of pFAK/total FAK, pSRC/total SRC, pPAX/total PAX etc. By visual inspection there seems to be a considerable lowering of pSRC while total SRC appears to be quite unchanged in the westerns provided

The manuscript will benefit if the authors address the above questions

Author Response

Reviewer 3

Cantharidin (and norcantharidin) are reported anti-cancer agents that potentially can escape multidrug resistance and therefore might be of interest as to establish therapeutic schemes alone or in combination with other better defined chemotherapeutic agents.

The authors conducted a research plan in line to similar studies and I only wish to note some seemingly conflicting results that the authors may wish to address in their response and the manuscript:

The dominant mechanism of CTD/NCTD is inhibitory activity against PP1 and PP2A protein phosphatases and this is largely ignored in the manuscript

è Thank you for your good suggestion. Based on your suggestion, we described it in the Introduction section like the below.

à “NCTD is known as a compound inhibitor of protein Ser/Thr phosphatases (PPP), which regulate various important physiological and pathological processes, such as cell cycle and proliferation (Ref. 1). [Lines 62-64]

Ref. 1 Li, Y.; Ge, Y.; Liu, F. Y.; Peng, Y. M.; Sun, L.; Li, J.; Chen, Q.; Sun, Y.; Ye, K., Norcantharidin, a protective therapeutic agent in renal tubulointerstitial fibrosis. Mol Cell Biochem 2012, 361, (1-2), 79-83.

Given the above I find it quite strange that phosphorylation of FAK (and even more of paxillin) is downregulated and that in this case is less dose sensitive than wound closure

è We fully understood what your comment is. As your commented, the activation of Paxillin (Y118) was more strongly attenuated by NCTD treatment than those of FAK (Y397). Despite the weakly regulated phosphorylation of FAK by NCTD treatment, it is difficult to fully explain the strong reduction of phosphorylation of Paxillin. However, given that phosphorylation of Paxillin is regulated by various stimuli and kinases such as growth factors and CAKb/Pyk2/CadTK, it is possible that NCTD may decrease phosphorylation of Paxillin via other unknown factors (Ref. 2). However, further research needs to examine detailed mechanism underlying that NCTD control phosphorylation of Paxillin in the future. Thank you for your kind and important comments again.

Ref. 2 Schaller, M. D., Paxillin: a focal adhesion-associated adaptor protein. Oncogene 2001, 20, (44), 6459-72.

ERK activation is a predominant feature of MEK aggresion and the authors should address the effects of NCTD on phospho-ERK levels

è We fully understand your comment. As you indicated, there are such scientific evidences about the functional link between NCTD-induced apoptosis and up-regulated phosphorylation of ERK. (Ref. 6-7) Thus, we performed western blotting on the expression of p-ERK, but NCTD exhibited no apparent effect on the expression of p-ERK. Thus, this result suggest that ERK activation is not related with NCTD-induced anticancer activty in our present study.

Ref. 6 Chen, Y. N.; Cheng, C. C.; Chen, J. C.; Tsauer, W.; Hsu, S. L., Norcantharidin-induced apoptosis is via the extracellular signal-regulated kinase and c-Jun-NH2-terminal kinase signaling pathways in human hepatoma HepG2 cells. Br J Pharmacol 2003, 140, (3), 461-70.

Ref.7 Yu, C. C.; Ko, F. Y.; Yu, C. S.; Lin, C. C.; Huang, Y. P.; Yang, J. S.; Lin, J. P.; Chung, J. G., Norcantharidin triggers cell death and DNA damage through S-phase arrest and ROS-modulated apoptotic pathways in TSGH 8301 human urinary bladder carcinoma cells. Int J Oncol 2012, 41, (3), 1050-60.

Activities of matrix metalloproteases 2 and 9 are differentially regulated but it will be important to show what is the trend of their overall amounts via westerns blotting. Additionally Fig. 2E lacks loading controls

è We totally understood what your comment is. As you suggested, we showed our loading controls for Figure 2E. Thus, we replaced old Figure 2E with new one. Thank you for your careful comment.

The very big difference in the inhibitory concentrations between NCTD and PF562271 further signifies that NCTD effects on FAK and paxillin phosphorylation maybe very indirect and thus the evaluation of NCTD without concrete knowledge of its target(s) is somewhat inconclusive

è We fully understand what your comment is. In our preliminary experiments, 0.5mM PF562271 treated YD-15 cells for 24 h altered the phosphorylation status of FAK without the cleavage of PARP. Thus, subsequent experiments were performed with low concentration of 0.5mM PF56227.Therefore, it is the difference in the inhibitory concentrations between NCTD and PF562271. As you suggested, the concentration of NCTD is quite high compared with that of PF562271, so it may be indirect or have offtarget effects. Thus, we need to perform further additional researches to fully address this issue in the future. Thank you for your kind and important comments again.

Phalloidin staining of actin is not immunofluorescence so this has to be changed in title 4.7 and any where else it is relevant

è Thank you for your good suggestion. We corrected it in the Materials and Methods section like the below.

à “4.7. F-actin staining using TRITC-conjugated phalloidin” [Line 301]

densitometric analysis of western blots should consider the ratios of pFAK/total FAK, pSRC/total SRC, pPAX/total PAX etc. By visual inspection there seems to be a considerable lowering of pSRC while total SRC appears to be quite unchanged in the westerns provided

è Thank you for your kind and thoughtful comments. As you suggested, we corrected it in Figure 3C and 4B. Also, triplicated western blotting results significantly (p<0.05) showed no changes in the expression of p-Src. Thank you for your kind comments again.